# Does seniority always correlate with simulated intubation performance? Comparing endotracheal intubation performance across medical students, residents, and physicians using a high-fidelity simulator

Sze-Yuen Yau[1], Yu-Che Chang[1,2,3], Meng-Yu Wu[1,3,4], Shu-Chen Liao [1,3,5]*

1 Chang Gung Medical Education Research Center (CGMERC), Chang Gung Memorial Hospital, Linkou, Taiwan, 2 Department of Emergency Medicine, Chang Gung Memorial Hospital, Linkou, Taiwan, 3 Chang Gung University College of Medicine, Taoyuan, Taiwan, 4 Department of Cardiac Surgery, Chang Gung Memorial Hospital, Linkou, Taiwan, 5 Department of Emergency Medicine, Keelung Chang Gung Memorial Hospital, Keelung, Taiwan

* ermdsusan@gmail.com

**Data Availability Statement:** All the raw data are available from the Figshare repositories (accession number: 10.6084/m9.figshare.14962944), URLs:

## Abstract

### Background

Endotracheal intubation is crucial in emergency medical care and anaesthesia. Our study employed a high-fidelity simulator to explore differences in intubation success rate and other skills between junior and senior physicians.

### Methods

We examined the performance of 50 subjects, including undergraduate students (UGY), postgraduate trainees (PGY), residents (R), and attending physicians (VS). Each participant performed 12 intubations (i.e. 3 devices x 4 scenarios) on a high-fidelity simulator. Main outcome measures included success rate, time for intubation, force applied on incisor and tongue, and Cormack Lehane grades.

### Results

There was no primary effect of seniority on any outcome measure except success rate and Cormack Lehane grades. However, VS demonstrated shorter duration than medical students using Glidescope and direct laryngoscopy, whereas VS and R applied significantly more force on the incisor in the normal airway and rigid neck scenario respectively.

### Discussion

Seniority does not always correlate with skill perfection in detailed processes. Our study suggests that the use of video laryngoscopy enhances the intubation success rate and

https://figshare.com/search?q=10.6084%2Fm9.
figshare.14962944).

**Funding:** In this study, Young Tah Instruments Ltd.
and Kyoto Kagaku Co.,Ltd. funded for borrowing
the simulator: MW13: DAM Simulator Training
Model for airway training and curriculum
development, but had no role in data analysis, data
interpretation, or manuscript writing. Otherwise,
there was no more funding.

**Competing interests:** In this study, Young Tah
Instruments Ltd. and Kyoto Kagaku Co.,Ltd. funded
for borrowing the simulator: MW13: DAM
Simulator Training Model for airway training and
curriculum development. This does not alter our
adherence to PLOS ONE policies on sharing data
and materials.

speed, but the benefit only accrues to senior learners, whereby they applied more force on
the incisor at a single peak under difficult scenarios. These findings are discussed in terms
of psychological and cognitive perspectives.

## Conclusion

Speed and safety are essential for high quality critical medical procedures. A tool should be
designed and implemented to educate junior physicians with an emphasis on practice and
efficiency, which should also contribute to updating senior physicians' knowledge and com-
petence by providing instant feedback on their performance. This type of fine-grained feed-
back could serve as a complement to traditional training and provide a sustainable learning
model for medical education.

## Introduction

Endotracheal intubation is considered a critical practice in emergency medical care, intensive
care medicine and anesthesia to maintain a secure and clear airway [1]. It is an invasive perfor-
mance that requires complex psychomotor skills especially in demand during the COVID-19
pandemic. The degree of success of intubation is traditionally evaluated by the time required
and the achievement of both-lung ventilation. Intubation may also be evaluated by additional
indicators such as applied force and glottic view that could affect patient outcome [2]. Profi-
cient physicians should be able to adjust their techniques according to patients' conditions to
maintain an airway and provide ventilation within a short period [3]; their force applied on
the incisor should be minimized during intubation with any technique [4–6], and they should
be able to obtain a good laryngeal view [7]. In general, the success of intubation is associated
with the patient's condition, intubation devices used, and physicians' level of competence [3].
A critical viewpoint has argued that the key contributor to success rate and intubation quality
is the operator's competence [8, 9]. Since seniority is one of the requirements for performing
difficult airway management, in cases where patients have difficult airway conditions, senior
physicians often play rescuing roles. Indeed, plenty of guidelines suggest involving senior per-
sonnel when making decisions, diagnoses, and when performing difficult intubations [9, 10].
Although previous retrospective studies to evaluate the impact of wearing personal protective
equipment during endotracheal intubation in the COVID-19 pandemic have reported that
seniority is correlated with intubation success rate [11], other studies have argued that senior-
ity is a poor predictor of expertise or proficiency [12, 13]. Expertise depends more on the vol-
ume of deliberate practice in which an individual has engaged.

In this context, we employed a high-fidelity simulator as our assessment model, and devel-
oped a difficult, simulated airway management course, incorporating different intubation
techniques to investigate the intubation performance. The main objective of this study is to
explore differences in intubation success rate and other skills during intubation under the
Kyoto Kagaku MW11: difficult airway management (DAM) simulator evaluation system.

## Methods

### Study design and context

We conducted a cross-sectional study to explore intubation performance across different
levels of seniority. The study protocol was approved by the Chang-Gung Medical

Foundation Institutional Review Board, and permission was granted (201802283B0C101) by the Medical Ethics Committee of Chang-Gung Memorial Hospital. In March 2018, we held a difficult airway intubation training course for medical students and physicians in the two branches of Chang-Gung Memorial Hospital, Linkou and Keelung (Fig 1). Eligible

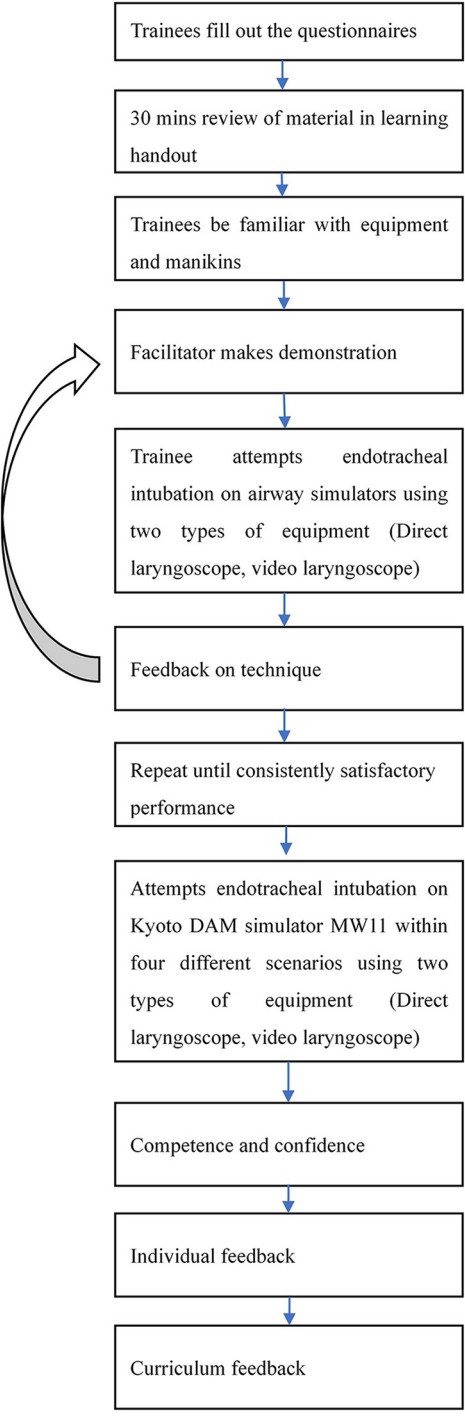

**Fig 1. Flowchart of steps in teaching and learning endotracheal intubation.**

participants included final-year medical students (UGY) and postgraduate year physicians (PGY) who rotate monthly through emergency department training, and residents (R) and attending physicians (VS) from the emergency department or surgery department. Formal medical education in Taiwan involves a seven-year undergraduate training (UGY), and students are required to take national licensing exams at the end of the 7th year. Licensed physicians are required to complete a one-year postgraduate training (PGY) for general medicine training and a four-year emergency medicine residency training (R) in order to become an attending physician (VS).

Written informed consent was obtained from the study participants, prior to participation, and all personal identifying information was kept confidential. We allocated a maximum teaching group size of four students to optimize teaching and learning quality. Each session lasted about 90 minutes and comprised 3 sections: 30-minute material review, 30-minute device familiarization (i.e., practice on direct laryngoscopy, Trachway®, and Glidescope®), and 30-minute manikin practice section (Fig 1). This design offered sufficient demonstration and practice time for learners to become familiar with the devices and intubation on the manikin but was not intended as an intervention. The simulators we used included "Laerdal® Airway Management Trainer and MW13: DAM Simulator Training Model". The training scenarios included difficult mouth openings, limitation of neck flexibility, and tongue swelling. We did not standardize the number of attempts during the 30-minute practice section, but instead gauged readiness according to learners' self-perception. At the end, each learner was assessed over 12 intubations (4 scenarios × 3 devices) that were always presented in the same order. Learners were assessed on a fixed sequence of four scenarios using direct laryngoscopy, Trachway®, and then Glidescope®. Four different levels of intubation scenarios were presented, from the easiest to the most difficult, as follows: (©Kyoto KagakuMW11: difficult airway management (DAM) simulator evaluation system).

1. Normal: Simulation of patient with no airway difficulties. Opening the mouth and retroflexion of the neck is the easiest of all four levels.

2. Locked jaw: Simulation of patient with locked jaw. Opening the mouth is more difficult.

3. Rigid neck: Simulation of patient with limitations in neck retroflexion. Users face difficulty when trying to raise the chin.

4. Micrognathia: Simulation of patient with micrognathism (recess of lower jaw). Opening the mouth as well as retroflexion of the neck are the most difficult.

## Outcome measures

We recorded the participants' first attempt performance for the analysis. The primary outcome measure was the success or failure of the intubation. Successful intubation was defined as follows: intubation time within 30s, and ventilation of both lungs. We recorded the total time needed for intubation in order to understand the learners' deficiencies. The secondary outcome measures of the learners' intubation skill included (i) maximum force applied on the incisor, (ii) maximum force applied on the tongue, and (iii) Cormack Lehane grades when passing the endotracheal tube. The Cormack Lehane grade analysis was limited to direct laryngoscopy because line-of-sight view from a video laryngoscope is from the camera lens of video chipsets, so Cormack Lehane grade differs from the original grading concept. Data was collected via ©Difficult Airway Management Simulator MW11 by ©Kyoto Kagaku.

## Statistical analysis

Statistical comparison was performed using a Chi-square test for the nominal dependent variable (i.e., success rate and Cormack Lehane grade) and a $4 \times 4 \times 3$ repeated measure ANOVA with one between-subject factor (i.e., seniority) and two within-subject factors (i.e., scenario and device) for continuous dependent variables (i.e., time to intubation and force applied on incisor and tongue) with post hoc Tukey's test via IBM® SPSS® Statistics Version 26. Follow up Tukey HSD for two factor ANOVA was performed with comparison among marginal means. Degree of freedom was corrected using Greenhouse-Geisser estimates if sphericity was lacking. All effects were reported as significant at $p < 0.05$.

Sample size was estimated for a Chi-square test and a repeated measure ANOVA using GPower 3.1 [14]. As there was no prior evidence for physician seniority, estimates were set based on a large effect size of 0.5 for the Chi-square test and 0.4 for repeated measure ANOVA, with a correlation among repeated measures of 0.3, an alpha of 0.05, and a statistical power of 0.8 [15]. A sample size of 44 for the Chi-square test and 28 for ANOVA is needed to test for appreciable differences between the four independent group means (physician seniority) across 12 intubations (4 scenarios × 3 devices).

## Results

Fifty-two learners were enrolled in the study, but two were subsequently excluded. One trainee did not accomplish the 12 intubations process, while another failed to provide informed consent. Data from the 50 individuals including medical students (UGY; n = 7), trainees (PGY; n = 18), residents (R; n = 18) and attending physicians (VS; n = 7) were collected for the analysis.

### Primary outcome: Intubation result

**(1) Success rate.** Based on the definition of intubation success, i.e., intubation time within 30s and ventilation of both lungs, the overall success rate was 44.7% (264/600). A chi-square test of independence indicated that success rate differed by seniority ($\chi^2_{(3)} = 15.1$, $p < 0.01$). VS (53/84, 63%) were more likely to achieve first pass success than UGY (36/84, 43%), PGY (96/216, 44%) and R (83/216, 38%).

**(2) Time of intubation.** The average time for intubation among learners was 44.3s ($SD = 38.2$s) (S1 Table). Surprisingly, the main effect of seniority was not significant ($F_{(3, 46)} = 1.89$, $p = 0.15$). However, significant interactions between seniority and device ($F_{(4.89, 74.90)} = 2.63$, $p < 0.05$) and between seniority and scenario ($F_{(5.88, 90.2)} = 3.00$, $p < 0.01$) were identified (Figs 2 & 3), and also the effect size was large (partial $\eta_p^2 = 0.15$ and partial $\eta_p^2 = 0.16$ respectively). VS ($m = 31.2$s, 95% CI 15.8–46.6; $m = 26.2$s, 95% CI 17.5–35.0) required significantly less time than UGY ($m = 71.6$s, 95% CI 56.2–87.0; $m = 50.6$s, 95% CI 41.8–59.3) when using direct laryngoscopy and Glidescope®, respectively ($p < 0.05$). However, no significant post hoc interaction between seniority and scenario was identified.

### Secondary outcome: Learners' intubation skills

**(1) Maximum applied force on incisor.** The mean force applied at a single peak was 52.7N ($SD = 50.7$N) (S1 Table). Among the twenty-five intubations (4.2%) in which at least 200 Newtons were applied, causing the front teeth to break, the distribution was: UGY (4/84; 4.8%), PGY (5/216; 2.3%), R (11/216; 5.1%), and VS (5/84; 6.0%), whereby VS represented the highest percentage. The interaction between seniority and scenario was significant ($F_{(6, 92)} = 2.30$, $p < 0.05$) with a medium effect size (partial $\eta_p^2 = 0.13$, Fig 4), but not the main effect of

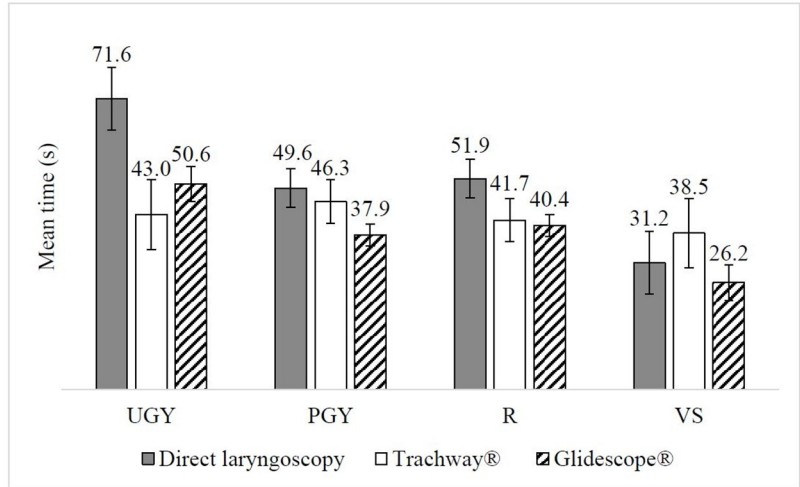

**Fig 2. Mean time to intubation(s) across UGY, PGY, R and VS using different intubation devices.** Error bars represent ±1 SE.

seniority ($F_{(3, 46)}$ = 1.72, $p$ = n.s.). VS ($m$ = 42.9N, 95% CI 31.2–54.6) applied significantly more force on the incisor compared to PGY ($m$ = 23.6N, 95% CI 16.3–30.9) in the normal airway, while R ($m$ = 74.7N, 95% CI 62.0–87.4) applied significantly more force compared to PGY ($m$ = 48.63N, 95% CI 35.9–61.3) in the rigid neck scenario ($p < 0.05$).

**(2) Maximum applied force on tongue.** The mean peak force applied on the tongue was 52.9N ($SD$ = 34.3N) (S1 Table). However, the main effect of seniority, and interactions were not significant ($F \leq 1.41$, $p$ = n.s).

**(3) Cormack Lehane grade when passing the endotracheal tube using direct laryngoscopy.** The majority of intubations were grade IV (94/200, 47%), followed by grade I (57/200,

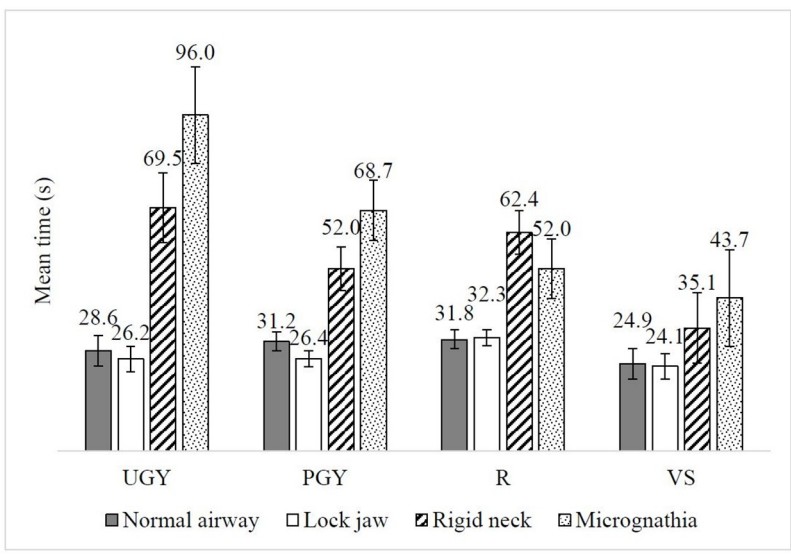

**Fig 3. Mean time to intubation(s) across UGY, PGY, R and VS in different intubation scenarios.** Error bars represent ±1 SE.

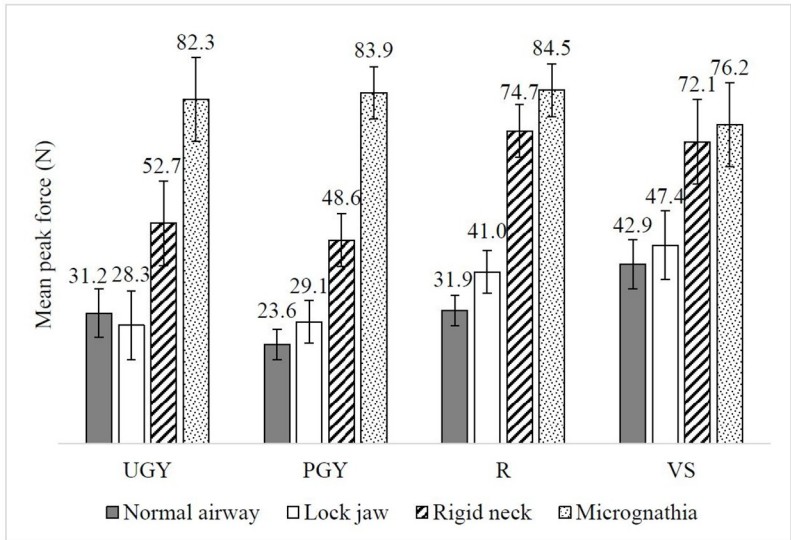

**Fig 4. Mean peak force (N) applied on incisors across UGY, PGY, R and VS in different intubation scenarios.**
Error bars represent ±1 SE.

**Table 1. Cormack Lehane grade across seniority using direct laryngoscopy in various scenarios.**

| | Seniority | | | |
|---|---|---|---|---|
| | UGY | PGY | R | VS |
| | (n = 28 intubations) | (n = 72 intubations) | (n = 72 intubations) | (n = 28 intubations) |
| Cormack Lehane grade | | | | |
| 1 | 11 (39.3%) | 20 (27.8%) | 16 (22.2%) | 10 (35.7%) |
| 2 | 5 (17.9%) | 8 (11.1%) | 7 (9.7%) | 4 (14.3%) |
| 3 | 3 (10.7%) | 9 (12.5%) | 9 (12.5%) | 4 (14.3%) |
| 4 | 9 (32.1%) | 35 (48.6%) | 40 (55.6%) | 10 (35.7%) |

28.5%) (Table 1). A chi-square test of independence indicated that Cormack Lehane grade did not differ by seniority ($\chi^2_{(9)}$ = 7.32, $p$ > 0.05).

## Discussion

The primary key finding of our study is that seniority correlates with success rate but not other skills. Given a 44% overall success rate across all seniorities, intubation appears quite difficult under the provided scenarios. It seems none of the groups are proficient enough to do well in these scenarios. Moreover, the non-significant main effect suggests that our learners' intubation speeds were not facilitated by increasing seniority, although previous study has reported a positive correlation [16, 17]. However, when intubation devices were taken into account, the significant interaction between seniority and device suggests that intubation performance was affected by different learners' seniority, but the effect depended on the devices used. The result shows that the use of video-laryngoscopy enhances the intubation speed of learners as reported in earlier study [18]. Indeed, previous experimental investigations have suggested that video-laryngoscopy provides a better laryngeal view and improves the first-attempt success rate and speed, compared to direct laryngoscopy [7, 19, 20]. It has also been highly recommended

during infectious, epidemic situations as a risk reduction method, such as during COVID-19 [21, 22]. However, our results show that junior learners remained unimproved.

By contrast, intubation skills were not affected by learners' seniority. There were no differences in the likelihood of obtaining the best glottis visualization when passing the endotracheal tube, across seniority. Moreover, the maximum applied force on incisor and tongue did not differ across seniority. The forces measured in the current study were much higher than in previous studies [4, 6, 23]. This could be explained by the use of different simulators and the sensitivity of the force measurement set up.

Although the mean peak force was substantially lower than the 200N that would cause front teeth breakage of normal incisors, we recorded 25 attempts by both juniors and seniors that reached this force, out of the 600 intubations in total. We cannot ignore the fact that the greatest number of these attempts with the greatest force on the incisors were by senior physicians. We also identified an increase of at least 9N with senior physicians (R and VS) compared to UGY and PGY. Moreover, increases of 11N applied on the tongue were recorded with seniors, compared to juniors. As we know, the glossopharyngeal nerve is positioned superior to the anterior surface of the epiglottis, and the vagus nerve has sensory pathways running distally from the posterior epiglottis to the lower airway. The pressure of laryngoscopy during endotracheal intubation will stimulate both nerves and may cause lethal cardiovascular responses [24, 25]. Therefore, minimizing the applied laryngoscopy force could reduce the haemodynamic response, as well as the local tissue trauma associated with intubation [26]. Accordingly, the laryngoscopy force applied to the soft tissues of the oral mucosa should be reduced as the physician's seniority and competence in the quality of their care increases. Although that increased level of force applied may not cause severe complications with patients' oral soft tissue, it does still represent a blemish on our senior physicians' intubation quality record. A recent study also reported the same finding that more experienced anesthetists generated higher peak dental forces during manikin laryngoscopy than less experienced individuals [6]. Thus, we employed a psychological perspective to understand the findings. Our results revealed that there may have been a speed-force trade-off: novice physicians focused on precision at a slower pace, whereas senior physicians emphasized speed with conditional accuracy. Experienced physicians might accept higher dental force in order to accomplish a difficult intubation, knowing that a failed intubation may risk life-threatening complications, such as hypoxemia and regurgitation or death, as compared to "mere" dental trauma [27, 28]. Previous qualitative study examining general emergency care practices has highlighted novice-expert differences in cognitive patterns [29]. Novice physicians rely on what has been learned and follow guidance and standard procedures; in this case, best visualization of the glottis and insertion within the appropriate range. Failures of novices are mostly associated with the lack of ability to solve the situation, and thus fewer successful attempts and less speed are expected. On the other hand, experienced physicians identify key inferences and integrate essential information into the diagnosis, leading to "pre-programming" from their extensive experience. This then produces fast and effortless intubation. However, performance is largely based on prior experience, and focus of attention, whereby strong habit intrusions may affect the diagnosis of rare medical cases, or some information and clinical steps that were missed if beyond their attention [30]. Previous survey-based national study has reported the same findings: more senior anesthesiologists achieved lower accuracy in grading the overall health status of a patient; i.e., more experienced personnel made judgements based on their clinical experiences rather than on the standards and criteria [31].

Having thus revealed some possible causes of senior physicians' greater dental force, other confounding factors should also be considered to improve performance. Our novel simulator analyzes different aspects of every attempt at intubation, and thus can characterize each user's

shortcomings and capabilities. Once the insufficiency is understood, skills can be refined. The possibility of complications or infection can be greatly reduced, which is especially significant during the COVID-19 pandemic.

## Limitations

Our study result may be affected by the self-selection effect. Learners who enrolled in the training course, especially juniors, may be, or may perceive themselves as, less competent or more motivated to learn. Therefore, measured intubation performance may not be generalizable to those who did not choose to enroll. Second, we did not measure objective data that could indicate whether the 4 groups (UGY, PGY, R, VS) differed in expertise with regard to any of the intubation techniques and manikins. Familiarity could have influenced their performance, which may explain medical students' superior performance. Moreover, we did not control the learners' number of attempts during the practice section and assessment; we only analyzed their first attempt at each scenario. Learners with a higher number of attempts may have benefitted from a practice effect and thereby improved their performance in later scenarios. Also, we did not deploy devices that are more specifically recommended with respect to the particular scenarios. According to the Difficult Airway Society 2015 guidelines for the management of unanticipated difficult intubation in adults, supraglottic airway device (SAD) or flexible fiberscope may be more suitable than laryngoscopy [10]. Furthermore, Kyoto-Kagaku MW11 difficult airway management simulator was developed at Waseda University in 2006 to serve as an active training device and as an advanced evaluation tool for surgical instruments and medical procedures [32, 33]. The simulator was built based on the concepts of a Japanese anesthesia textbook (Tracheal Intubation Visual Manual of Clinical Basic Techniques) [34]. However, we were unable to discover any English literature describing the process of Cormack and Lehane grade verification [35]. Lastly, we may have overlooked the association between professional experience and intubation experience, and they may not always be positively correlated [36]. Senior learners may actually have less intubation experience than certain juniors.

## Conclusion

Speed and safety are essential for emergency and critical care. We should emphasize high quality and precision of critical medical procedures along with speed (short intubation time) as the indicators of satisfactory outcome, especially in the COVID-19 pandemic.

Although our study is only a small-scale investigation, it demonstrates the use of a high-fidelity simulator to deepen our understanding of intubation quality. To the best of our knowledge, our study is the first to incorporate all intubation variables (i.e., physicians, devices used, and intubation scenarios) into the investigation. A comprehensive training program encompassing knowledge, clinical skills and capabilities is necessary for the education of junior physicians, while senior physicians should be aware of habit intrusions that have been "pre-programmed" from their extensive experiences; effective and regular learning opportunities contribute to the updating of knowledge and refreshing of skills. A tool that emphasizes practice and efficiency should be designed and implemented to educate junior physicians, which could also contribute to the updating of senior physicians' knowledge and competence by providing instant feedback on their performance. Such fine-grained feedback could serve as a complement to traditional training and provide a sustainable learning model for medical education.

## Supporting information

**S1 Table. Time for intubation attempts and peak force applied on incisor and tongue across seniority using different devices in various scenarios.** Mean and 95% CI are presented.
(DOCX)

## Acknowledgments

The authors would like to thank all trainers and trainees who contributed to this study. We would like to express our thanks to Mr. Joji Araki for the consultation for the simulator repair and debugging. We would like to thank Professor Akihiko Kawauraa for his consultation in statistical analysis and constructive opinions for this manuscript.

## Author Contributions

**Conceptualization:** Yu-Che Chang.

**Data curation:** Meng-Yu Wu, Shu-Chen Liao.

**Formal analysis:** Meng-Yu Wu.

**Investigation:** Shu-Chen Liao.

**Methodology:** Sze-Yuen Yau, Shu-Chen Liao.

**Project administration:** Shu-Chen Liao.

**Software:** Meng-Yu Wu.

**Supervision:** Shu-Chen Liao.

**Writing – original draft:** Sze-Yuen Yau.

**Writing – review & editing:** Yu-Che Chang, Shu-Chen Liao.

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
