## [Decision Letter · Decision Letter 0]

20 May 2021

PONE-D-21-11478

Does seniority always correlate with proficiency? Comparing endotracheal intubation performance across medical students, PGY and physicians using a high-fidelity simulator

PLOS ONE

Dear Dr. Liao,

Thank you for submitting your manuscript to PLOS ONE. After careful consideration, we feel that it has merit but does not fully meet PLOS ONE’s publication criteria as it currently stands. Therefore, we invite you to submit a revised version of the manuscript that addresses the points raised during the review process.

Publication of this paper would depend on whether you can explain and justify your statistical methods and whether you can build a case that the study findings have significant implications for assessment of intubation proficiency with a mannequin.

We look forward to receiving your revised manuscript.

Kind regards,

Laura Pasin

Academic Editor

PLOS ONE

Journal Requirements:

2)  Please provide additional details regarding participant consent. In the ethics statement in the Methods and online submission information, please ensure that you have specified (1) whether consent was informed and (2) what type you obtained (for instance, written or verbal, and if verbal, how it was documented and witnessed). If your study included minors, state whether you obtained consent from parents or guardians. If the need for consent was waived by the ethics committee, please include this information.

3) Thank you for stating the following in the Acknowledgments Section of your manuscript:

[The authors wish to acknowledge Young Tah Instruments Ltd. and Kyoto Kagaku Co.,Ltd. for generously borrowing us the Difficult Airway Management Simulator (DSAM) MW11 and helping us to complete the research smoothly]

 [The author(s) received no specific funding for this work.]

Additionally, because some of your funding information pertains to commercial funding, we ask you to provide an updated Competing Interests statement, declaring all sources of commercial funding.

In your Competing Interests statement, please confirm that your commercial funding does not alter your adherence to PLOS ONE Editorial policies and criteria by including the following statement: "This does not alter our adherence to PLOS ONE policies on sharing data and materials.” as detailed online in our guide for authors  http://journals.plos.org/plosone/s/competing-interests.  If this statement is not true and your adherence to PLOS policies on sharing data and materials is altered, please explain how.

Please include the updated Competing Interests Statement and Funding Statement in your cover letter. We will change the online submission form on your behalf.

4) Please amend your manuscript to include your abstract after the title page.

5) Please include captions for your Supporting Information files at the end of your manuscript, and update any in-text citations to match accordingly. Please see our Supporting Information guidelines for more information: http://journals.plos.org/plosone/s/supporting-information.

Reviewers' comments:

Reviewer's Responses to Questions

**Comments to the Author**

1. Is the manuscript technically sound, and do the data support the conclusions?

Reviewer #1: Partly

Reviewer #2: Yes

2. Has the statistical analysis been performed appropriately and rigorously? 

Reviewer #1: Yes

Reviewer #2: I Don't Know

3. Have the authors made all data underlying the findings in their manuscript fully available?

Reviewer #1: Yes

Reviewer #2: No

4. Is the manuscript presented in an intelligible fashion and written in standard English?

Reviewer #1: Yes

Reviewer #2: Yes

5. Review Comments to the Author

Reviewer #1: Introduction is too long. Most introduction should be moved in the discussion section. Please try to focus on the main problem, what is known and your aims.

In particular last sentence of the introduction should state: - Your main objective and your secondary objectives.

methods

Please every acronym should be explained in full at first appearence.

£We quantified intubation skills into (1) maximum applied

121 force on incisor, (2) maximum applied force on tongue, and (3) Cormack Lehane grades

when passing the endotracheal tube." why these should quantify inyubation skills? An excessive pressure on incisor is avoidable but a low force will result in undervision. How do you determine the best force to be applied?Moreover,CL grades should not be related to the operator but to the patient. I believe you should analyze number of participants that assign a wrong CL grade.

why do you not use time to perform intubation?

Statistical analysis

"Sample size was estimated using GPower 3.1 to evaluate the minimum sample size

125 required using a two-tailed test, a medium effect size (d = 0.5), and an alpha of 0.05. " Required for what? Please state the outcome, the meand and the standard deviation (and the relative source) you used for power calculation.

success rate is not a continuous variable

Results

Two participants were excluded

"because they did not complete the final assessment as instructed" what does it mean?

Discussion

Other studies investaigated proficiency and experience (ie "Assignment of ASA-physical status relates to anesthesiologists' experience: a survey-based national-study. Korean J Anesthesiol. 2019 Feb;72(1):53-59. doi: 10.4097/kja.d.18.00224. Epub 2018 Nov 14. PMID: 30424587; PMCID: PMC6369346.") Please discuss your results at the light of the other studies.

Reviewer #2: Important educational Issues:

The authors address the topic of whether provider seniority correlates with performance of endotracheal intubation on a high fidelity manikin that allows 4 different scenarios. From an educational perspective, the findings would be valuable if they established whether performance correlates across different manikin scenarios or across different intubation techniques? The study does not address another important topic, how performance in manikins would relate to performance in patients.

Major Issues

1. Lines 123-135. Please provide more information on the statistical analysis. Did you perform the analysis with a repeated measures ANOVA model divided across technique and scenario? How did you account for seniority? Was the power analysis constructed for the same model used in the statistical analysis? The following two questions also indicate my confusion over the statistical analysis was performed

2. Success rates are expressed as fractions with error bars in Figure 2 and Supplementary Table 1. Were means and standard deviations calculated from an individual subject’s success on each intubation expressed as a binary variable (0 or 1)? Only 1 measurement of success or failure per subject was available for each of the technique-scenario combinations in Supplementary Table 1, yielding a binary result. A continuous estimate of success rate could only be achieved for the group (sum of successes divided by group n) with no standard deviation available. Since individual success was a binary variable, a mixed effect multivariate logistic regression accounting for clustering across seniority might be a more appropriate statistical test.

3. How did you calculate means and error bars for each combination of seniority and technique in Figures 2-3. Did you calculate a success rate and mean intubation duration for each subject that combined values from the 4 scenarios? Similarly, did you average subject results across the 3 techniques for Figures 4 and 5? Was averaging across techniques or scenarios part of the statistical analysis?

4. Supplementary Table 1 contains processed data. Other investigators could do more consequential analyses if the raw data were provided, i.e. results for each subject for all 4 scenarios, the 3 techniques, and labeled by subject seniority group.

5. The introduction should explain why the research described in the manuscript was important. What was the clinical, educational, or scientific value of evaluating performance of endotracheal intubation on mannequins as a function of operator seniority?

6. Furthermore, the introduction should also state the goal of the study and list the hypotheses. For example, did you expect performance to improve with seniority? Or, did you anticipate how scenario or technique might affect performance.

7. Why did you compare subjects’ laryngoscopy performance on the basis of seniority, rather than experience. According to Anders Ericsson (Med Educ 2007; 41:1124), seniority is a poor predictor of expertise or proficiency. Expertise depends more on the volume of deliberate practice in which an individual has engaged. For endotracheal intubation, the number of procedures the person has performed would be a better estimate than seniority.

8. Other studies have compared laryngoscopy performance measures among operators at different levels. Kerrey et al. Simul Healthc 2020; 15:251, Garcia et al. Br J Anaesth 2015; 115:302, and Hastings et al. Simul Healthc Dec 2 2020; PMID: 33273420 are just a few examples. The introduction or discussion could summarize appropriate studies from the literature to elucidate what is novel about the authors’ research project.

9. Line 140. The paper currently provides no objective data to indicate whether the 4 groups (UGY, PGY, R, VS) differ in expertise with any of the intubation techniques. Do the authors have information about the number of times subjects had performed endotracheal intubation in patients, manikins, or both? In particular, information on experience with any of the Kyoto Kagaku DAM System scenarios would help establish whether the groups could be expected to differ in expertise and performance under the research conditions. The data could also be included as covariate(s) in the analysis.

10. The results of a test will not differ between more proficient and less proficient individuals if the test is so difficult that everybody performs poorly. The same is true if the test is extremely easy and everybody does very well. Intubation appears quite difficult on all 4 Kyoto-Kagaku DAM System scenarios given the 44% success rate across all seniorities, the low success rate in the most senior group, and the high incidence, well over 50%, of grade 3 and grade 4 laryngeal views. An alternative explanation would be that none of the groups are proficient enough to do well on these scenarios. No data are available in the current version of the manuscript to make a conjecture one way or the other. Finally, the tests may not differ between groups because they do not actually reflect proficiency. Again, we do not have enough data to decide one way or the other. I think these points would be worth including in the discussion.

Minor Issues

11. Line 60-61. The statement that accurate force measurement has been unavailable until now is misleading. Laryngoscopy forces have been measured and studied for 25 years or more (Bishop et al., Anesth Analg 1992 74:411-4 , Hastings et al, Anesth Analg 1996; 82:462-8) and studies are ongoing to this day.

12. Line 78-79. I don’t think one can make the statement that there is no difference in skill acquisition and retention of laryngoscopy skills between biologic models and simulation models. Skill transfer is problematic with practice on manikins and it has not been demonstrated that an individual can develop the level of skill necessary for laryngoscopy in patients solely by practicing in a simulated environment. Novices must practice on patients to become proficient.

13. Lines 98-100. The Methods should specify what manikins and scenarios were used for practice, how many attempts were made by each subject, and how it was determined that practice time was sufficient.

14. Line 114. The statement, “We recorded their first attempt performance for the analysis” implies that subjects attempted a technique more than once on each scenario, even though only the first attempt was analyzed. How many attempts were made? Why were multiple attempts allowed if only the first was analyzed?

Line 139. Could you provide more information about how individuals in the PGY and R groups differed. Were the PGY trainees in their first year after medical school graduation, a position known as “intern” or PGY-1 in the US. Did the R group correspond to the level known as “resident” in the US?

15. Tables 2-5 should have legends explaining what the columns and error bars represent and how they were calculated.

16. Lines 179-184 and Table 1. To my knowledge, the Kyoto-Kagaku DAM System uses an algorithm based on the manikin head and neck geometry to estimate the laryngeal view. In this study population, a high percentage of the intubations were characterized by poor views, even with the Glidescope. It would be worthwhile for the investigators to determine whether the DAM laryngeal view estimate agreed with the view determined by an expert operator over the different techniques and scenarios.

17. Line 214. The Schieren reference, #38, reports that more experienced anesthetists actually generated higher peak dental forces during manikin laryngoscopy than did less experienced individuals. This is consistent with your finding. Experienced operators might accept dental force in order to accomplish a difficult intubation, realizing that the a failed intubation may risk more serious morbidity or death compared to dental trauma.

18. The 90-minute airway course and intubating testing involved a considerable number of procedures. Could fatigue have affected results, especially in the second half of the study with intubations requiring higher peak forces and longer durations.

19. Did the seniority groups differ in the amount of practice on any of the scenarios during the airway course? Could superior results by the UGY group on some scenarios reflect greater practice under those conditions in the course or prior to the study?

6. PLOS authors have the option to publish the peer review history of their article (what does this mean?). If published, this will include your full peer review and any attached files.

Reviewer #1: No

Reviewer #2: No

---

## [Author Response · Author response to Decision Letter 0]

14 Jul 2021

Reviewer #1: 

Introduction is too long. Most introduction should be moved in the discussion section. Please try to focus on the main problem, what is known and your aims.

In particular last sentence of the introduction should state: - Your main objective and your secondary objectives.

Our Response:

We thank the reviewer for this comment – and we have now tried to simplify our introduction. Up front, we state the objectives of the study as follows: 

“The main objective of this study is to explore differences in intubation success rate and skill proficiency across different levels of seniority under various conditions, and to gain insights into invisible risks and deficiencies.”

Reviewer #1: methods

Please every acronym should be explained in full at first appearance. “We quantified intubation skills into (1) maximum applied force on incisor, (2) maximum applied force on tongue, and (3) Cormack Lehane grades when passing the endotracheal tube." why these should quantify intubation skills? An excessive pressure on incisor is avoidable but a low force will result in undervision. How do you determine the best force to be applied? Moreover, CL grades should not be related to the operator but to the patient. I believe you should analyze number of participants that assign a wrong CL grade. why do you not use time to perform intubation?

Our Response:

With respect, we do take “time to perform intubation” as our primary outcome measure of intubation performance in the primary analysis. The maximum applied force on incisor and tongue, and Cormack Lehane grades when passing the endotracheal tube are additional secondary outcome measures to better understand performance. According to [Takeuchi S. et al. Longitudinal acquisition of endotracheal intubation skills in novice physicians. PLoS One. 2017;14;12(11):e0188224; GaudioRM,et al. Traumatic dental injuries during anaesthesia: part I: clinical evaluation. Dent Traumatol. 2010;26(6):459-65; Quinn JB, et al. Schieren M. et al. Comparison of forces acting on maxillary incisors during tracheal intubation with different laryngoscopy techniques: a blinded manikin study. Anaesthesia. 2019;74(12):1563-1571.], force applied on the incisor should be minimized during intubation with any techniques. Therefore, we take “applied force” as our secondary outcome to assess an individual’s intubation performance. Each participant must respond to the same scenarios with the same techniques, while the different CL grades achieved across the 12 intubations indicate their ability to. We appreciate the reviewer’s comment and will keep it in mind for the analysis of failed intubations. We have now rephrased the sentence and expanded the rationale behind our decision.

Reviewer #1: Statistical analysis

"Sample size was estimated using GPower 3.1 to evaluate the minimum sample size required using a two-tailed test, a medium effect size (d = 0.5), and an alpha of 0.05. " Required for what? Please state the outcome, the mean and the standard deviation (and the relative source) you used for power calculation.

success rate is not a continuous variable

Our Response:

We thank the reviewer for this comment. We have now refined the description of the sample size estimation that fit the same model used in the statistical analysis as follows:

“Sample size was estimated for a Chi-square test and a repeated measure ANOVA using GPower 3.1. As there was no prior evidence for physician seniority, estimates were set based on a large effect size of 0.5 for the Chi-square test and 0.4 for repeated measure ANOVA, with a correlation among repeated measures of 0.3, an alpha of 0.05, and a statistical power of 0.8. A sample size of 44 for the Chi-square test and 28 for ANOVA is needed to test for appreciable differences between the four independent group means (physician seniority) across 12 intubations (4 scenarios × 3 devices).”

We agree that the individual subject’s success rate on each intubation should be a binary variable and we have modified it in our revised version.

Reviewer #1: Results

Two participants were excluded “because they did not complete the final assessment as instructed” what does it mean?

Our Response:

We have now re-phrased the sentence to be more explicit:

“Fifty-two learners were enrolled in the study, but two were subsequently excluded. One trainee did not accomplish the 12 intubations process, while another failed to provide informed consent.”

Reviewer #1: Discussion

Other studies investigated proficiency and experience [i.e., "Assignment of ASA-physical status relates to anesthesiologists' experience: a survey-based national-study. Korean J Anesthesiol. 2019 Feb;72(1):53-59. doi: 10.4097/kja.d.18.00224. Epub 2018 Nov 14. PMID: 30424587; PMCID: PMC6369346."] Please discuss your results at the light of the other studies.

Our Response:

Thank you for the comments– this study reported the same finding, i.e., that experienced personnel tended to rely on their clinical experience (habit intrusions) more than the actual evidence when grading patients. We have discussed this study and some other appropriate studies in the discussion section.

Reviewer #2: Important educational Issues:

The authors address the topic of whether provider seniority correlates with performance of endotracheal intubation on a high-fidelity manikin that allows 4 different scenarios. From an educational perspective, the findings would be valuable if they established whether performance correlates across different manikin scenarios or across different intubation techniques? The study does not address another important topic, how performance in manikins would relate to performance in patients.

Our Response:

We thank the reviewer for the comments – the main focus of this study is to explore differences in intubation success rate and skill proficiency across different levels of seniority under various conditions and to gain insights into invisible risks and deficiencies, rather than to design and implement a simulation program. We have revised our introduction, discussion and conclusion sections to tie in better with our objective.

Previous study has reported that paramedics trained in endotracheal intubation based on a systematic manikin-only teaching program can attain acceptable individual success rates in actual field settings [Stratton SJ. Et al. Prospective study of manikin-only versus manikin and human subject endotracheal intubation training of paramedics. Ann Emerg Med. 1991;20 (12):1314-8.]. However, we do agree that a single study is not sufficient to document how performance on manikins would relate to performance on patients. Additionally, we have rephrased our conclusion section to open discussion of educational intervention.

Reviewer #2: Major Issues

1. Lines 123-135. Please provide more information on the statistical analysis. Did you perform the analysis with a repeated measures ANOVA model divided across technique and scenario? How did you account for seniority? Was the power analysis constructed for the same model used in the statistical analysis? The following two questions also indicate my confusion over the statistical analysis was performed

Our Response:

We have now refined the description of the statistical analysis to be more explicit and we have updated our power analysis to fit the same model used in the statistical analysis:

“Statistical comparison was performed using a Chi-square test for the nominal dependent variable (i.e., success rate and Cormack Lehane grade) and a 4 × 4 × 3 repeated measure ANOVA with one between-subject factor (i.e., seniority) and two within-subject factors (i.e., scenario and device) for continuous dependent variables (i.e., time to intubation and force applied on incisor and tongue) with post hoc Tukey's test via IBM® SPSS® Statistics Version 26. Follow up Tukey HSD for two factor ANOVA was performed with comparison among marginal means. Degree of freedom was corrected using Greenhouse-Geisser estimates if sphericity was lacking. All effects were reported as significant at p< 0.05.”

Physician seniority was defined as follows: medical students (final-year medical students of the formal seven-year medical school), PGY (postgraduate year one, the first year of graduate training after medical school, focusing on general medical training), residents (participants in a 4-year residency training program in emergency medicine developed by the Taiwan Society of Emergency Medicine, who have completed their one-year PGY program), and attending physicians (who have completed the residency training program in an accredited teaching hospital and are board certified in emergency medicine or surgery). These details have been added in the revised manuscript.

Reviewer #2:

2. Success rates are expressed as fractions with error bars in Figure 2 and Supplementary Table 1. Were means and standard deviations calculated from an individual subject’s success on each intubation expressed as a binary variable (0 or 1)? Only 1 measurement of success or failure per subject was available for each of the technique-scenario combinations in Supplementary Table 1, yielding a binary result. A continuous estimate of success rate could only be achieved for the group (sum of successes divided by group n) with no standard deviation available. Since individual success was a binary variable, a mixed effect multivariate logistic regression accounting for clustering across seniority might be a more appropriate statistical test.

Our Response:

We thank the reviewer for this comment – and we agree that an individual participant’s success on each intubation should be expressed as a binary variable. As the main focus of the study is the difference in intubation proficiency across different levels of seniority, we decided to use a simple method – Chi square test – to identify the difference. However, we appreciate the reviewer’s recommendation and retain it for our further investigation. 

Reviewer #2:

3. How did you calculate means and error bars for each combination of seniority and technique in Figures 2-3. Did you calculate a success rate and mean intubation duration for each subject that combined values from the 4 scenarios? Similarly, did you average subject results across the 3 techniques for Figures 4 and 5? Was averaging across techniques or scenarios part of the statistical analysis?

Our Response:

Yes, the mean and error bars (i.e., ±1 SE) were calculated for each significant interaction of repeated measure ANOVA - we have added the details in the statistical analysis mentioned above. 

Reviewer #2:

4. Supplementary Table 1 contains processed data. Other investigators could do more consequential analyses if the raw data were provided, i.e. results for each subject for all 4 scenarios, the 3 techniques, and labeled by subject seniority group.

Our Response:

We will gladly provide our raw data. 

Reviewer #2:

5. The introduction should explain why the research described in the manuscript was important. What was the clinical, educational, or scientific value of evaluating performance of endotracheal intubation on mannequins as a function of operator seniority?

Our Response:

Thank you - we have now revised our introduction to explain the rationale of the research more clearly: 

“Since seniority is one of the requirements for performing difficult airway management, in cases where patients have difficult airway conditions, senior physicians often play rescuing roles. Indeed, plenty of guidelines suggest involving senior personnel when making decisions, diagnoses, and when performing difficult intubations. Although previous retrospective studies to evaluate the impact of wearing personal protective equipment during endotracheal intubation in the COVID-19 pandemic have reported that seniority is correlated with intubation success rate, other studies have argued that seniority is a poor predictor of expertise or proficiency. Expertise depends more on the volume of deliberate practice in which an individual has engaged.”

Reviewer #2:

6. Furthermore, the introduction should also state the goal of the study and list the hypotheses. For example, did you expect performance to improve with seniority? Or, did you anticipate how scenario or technique might affect performance.

Our Response:

As mentioned above - we could not anticipate whether performance would improve with seniority. Therefore, we needed to design a study to explore their performance under various conditions (i.e., managing difficult airways using difficult devices), and to gain insights into invisible risks and deficiencies.

Reviewer #2:

7. Why did you compare subjects’ laryngoscopy performance on the basis of seniority, rather than experience. According to Anders Ericsson (Med Educ 2007; 41:1124), seniority is a poor predictor of expertise or proficiency. Expertise depends more on the volume of deliberate practice in which an individual has engaged. For endotracheal intubation, the number of procedures the person has performed would be a better estimate than seniority.

Our Response:

Thank you – and this is the reason we designed this study, as mentioned above. 

“Since seniority is one of the requirements for performing difficult airway management, in cases where patients have difficult airway conditions, senior physicians often play rescuing roles. Indeed, plenty of guidelines suggest involving senior personnel when making decisions, diagnoses, and when performing difficult intubations. Although previous retrospective studies to evaluate the impact of wearing personal protective equipment during endotracheal intubation in the COVID-19 pandemic have reported that seniority is correlated with intubation success rate, other studies have argued that seniority is a poor predictor of expertise or proficiency. Expertise depends more on the volume of deliberate practice in which an individual has engaged.”

We appreciate the idea of “number of procedures” in determining skill proficiency and we are now designing a possible study to measure the physical and cognitive impact.

Reviewer #2:

8. Other studies have compared laryngoscopy performance measures among operators at different levels. Kerrey et al. Simul Healthc 2020; 15:251, Garcia et al. Br J Anaesth 2015; 115:302, and Hastings et al. Simul Healthc Dec 2 2020; PMID: 33273420 are just a few examples. The introduction or discussion could summarize appropriate studies from the literature to elucidate what is novel about the authors’ research project.

Our Response:

Thank you. We have summarized appropriate studies from the literature. 

Reviewer #2:

9. Line 140. The paper currently provides no objective data to indicate whether the 4 groups (UGY, PGY, R, VS) differ in expertise with any of the intubation techniques. Do the authors have information about the number of times subjects had performed endotracheal intubation in patients, manikins, or both? In particular, information on experience with any of the Kyoto Kagaku DAM System scenarios would help establish whether the groups could be expected to differ in expertise and performance under the research conditions. The data could also be included as covariate(s) in the analysis.

Our Response:

From our hospital records, the number of PGY physicians who performed tracheal intubation was about 10 cases each year. Emergency residents performed tracheal intubation in about 50-90 cases each year, with different resident seniority. Attending physicians performed 20-30 cases each year as a seniority rescue in difficult airway cases. Since medical student trainees have no physician licenses, there are very few opportunities to perform this invasive procedure except during their anesthesia training course. Regarding the experience with manikins, UGY had more practice time (about once each month) on manikins. Final year medical students have more experience on manikins because they need to pass the Medical Licensing Examination, and airway management is one of the core competencies. PGY have some regular airway training courses, and practice time on manikins is 3-5 times/ year. Residents and attending physicians have very little practice opportunity on manikins unless they are on the faculty of respiratory training courses.

Reviewer #2:

10. The results of a test will not differ between more proficient and less proficient individuals if the test is so difficult that everybody performs poorly. The same is true if the test is extremely easy and everybody does very well. Intubation appears quite difficult on all 4 Kyoto-Kagaku DAM System scenarios given the 44% success rate across all seniorities, the low success rate in the most senior group, and the high incidence, well over 50%, of grade 3 and grade 4 laryngeal views. An alternative explanation would be that none of the groups are proficient enough to do well on these scenarios. No data are available in the current version of the manuscript to make a conjecture one way or the other. Finally, the tests may not differ between groups because they do not actually reflect proficiency. Again, we do not have enough data to decide one way or the other. I think these points would be worth including in the discussion.

Our Response:

Thank you for your comments. These points are worthy of being included in the discussion. 

Reviewer #2: Minor Issues

11. Line 60-61. The statement that accurate force measurement has been unavailable until now is misleading. Laryngoscopy forces have been measured and studied for 25 years or more (Bishop et al., AnesthAnalg 1992 74:411-4 , Hastings et al, AnesthAnalg 1996; 82:462-8) and studies are ongoing to this day.

Our Response:

We agree and the statement has been removed in the revision. 

Reviewer #2:

12. Line 78-79. I don’t think one can make the statement that there is no difference in skill acquisition and retention of laryngoscopy skills between biologic models and simulation models. Skill transfer is problematic with practice on manikins and it has not been demonstrated that an individual can develop the level of skill necessary for laryngoscopy in patients solely by practicing in a simulated environment. Novices must practice on patients to become proficient.

Our Response:

We thank the reviewer for the comment and have now removed this controversial statement. 

Reviewer #2:

13. Lines 98-100. The Methods should specify what manikins and scenarios were used for practice, how many attempts were made by each subject, and how it was determined that practice time was sufficient.

Our Response:

Thank you, the simulators we used included“ Laerdal® Airway Management Trainer and MW13: DAM Simulator Training Model”; the training scenarios included difficult mouth openings, limitation of neck flexibility, and tongue swelling. We did not standardize the practice time before the final assessment but left it to the learners’ discretion, based on their own self-perceived readiness instead. We have now added these details into the method section.

Reviewer #2:

14. Line 114. The statement, “We recorded their first attempt performance for the analysis” implies that subjects attempted a technique more than once on each scenario, even though only the first attempt was analyzed. How many attempts were made? Why were multiple attempts allowed if only the first was analyzed?

Our Response:

We did not control the number of attempts as the study context was originally designed as a training course. Therefore, we only analyzed their first attempt performance as each learner was requested to perform at least once with a given technique under each scenario. Indeed, learners making multiple attempts may experience a practice effect and improve their performance under later scenarios. We have now added this issue to our limitations section. 

Reviewer #2:

Line 139. Could you provide more information about how individuals in the PGY and R groups differed. Were the PGY trainees in their first year after medical school graduation, a position known as “intern” or PGY-1 in the US. Did the R group correspond to the level known as “resident” in the US?

Our Response: (refer to Q1)

It is not equivalent to the US system. In Taiwan, we established the 1-year PGY training program in 2011, focusing on general medicine training. PGY refers to the first year of graduate training after completing the formal seven years of medical school and getting a physician’s license. During the one-year PGY training, physicians receive four months’ internal medicine training, two months’ general surgery training, one month of emergency medicine training, one month each of pediatric- and obstetrics/gynecology training, two months in community medicine, and one month of self-selected training. After completing the PGY training course, they can apply for a resident program, which refers to a subsequent 4-6 years of residency training in different specialties.

Reviewer #2:

15. Tables 2-5 should have legends explaining what the columns and error bars represent and how they were calculated.

Our Response:

Thank you. We have now added legends to explain what the error bars represent (i.e., error bars represent ±1 SE).

Reviewer #2:

16. Lines 179-184 and Table 1. To my knowledge, the Kyoto-Kagaku DAM System uses an algorithm based on the manikin head and neck geometry to estimate the laryngeal view. In this study population, a high percentage of the intubations were characterized by poor views, even with the Glidescope. It would be worthwhile for the investigators to determine whether the DAM laryngeal view estimate agreed with the view determined by an expert operator over the different techniques and scenarios.

Our Response:

There are four scenarios in the Kyoto-Kagaku DAM System, including normal, locked jaw, rigid neck, and micrognathia. In the rigid neck scenario, the degree of neck stiffness is similar to patients with ankylosing spondylitis or patients wearing cervical collars when using direct or video laryngoscopy to perform endotracheal intubation. In the micrognathia scenario, the difficulty of tracheal management is similar to patients who received radiotherapy or who have had long-term use of betel nut, which leads to muscle fibrosis. In the rigid neck and micrognathia scenarios, instead of laryngoscopy, supraglottic airway device (SAD) or flexible fiberscope may be more suitable devices to perform intubation [Frerk, C. et al. Difficult Airway Society 2015 guidelines for management of unanticipated difficult intubation in adults. BJA: British Journal of Anaesthesia, 2015, 115(6), 827-848.]. One limitation of our study is that we should consider providing more devices like supraglottic airway device (SAD) or flexible fiberscope in future studies and provide more training following the Difficult Airway Society guidelines. 

Reviewer #2:

17. Line 214. The Schieren reference, #38, reports that more experienced anesthetists actually generated higher peak dental forces during manikin laryngoscopy than did less experienced individuals. This is consistent with your finding. Experienced operators might accept dental force in order to accomplish a difficult intubation, realizing that the a failed intubation may risk more serious morbidity or death compared to dental trauma.

Our Response:

Thank you - we have now re-phrased the sentence to be more explicit and added these comments in the discussion. 

Reviewer #2:

18. The 90-minute airway course and intubating testing involved a considerable number of procedures. Could fatigue have affected results, especially in the second half of the study with intubations requiring higher peak forces and longer durations.

Our Response:

We agree that fatigue could degrade learners’ performance and we have added this issue to our limitations section. Although we did not receive any complaints from learners, we could measure learners’ fatigue levels to provide a reference point for the training course design. 

Reviewer #2:

19. Did the seniority groups differ in the amount of practice on any of the scenarios during the airway course? Could superior results by the UGY group on some scenarios reflect greater practice under those conditions in the course or prior to the study?

Our Response:

As mentioned above, we did not standardize the participants’ practice time – and we did not record their practice time. We have included this issue in our limitations section as practice might have an impact on the participants’ performance in the final assessment. Moreover, we agree that familiarity with manikins could be a possible explanation behind UGY’s superior results, and we have added this to our discussion section.

---

## [Decision Letter · Decision Letter 1]

18 Aug 2021

PONE-D-21-11478R1

Does seniority always correlate with proficiency? Comparing endotracheal intubation performance across medical students, PGY and physicians using a high-fidelity simulator

PLOS ONE

Dear Dr. Liao,

Thank you for submitting your manuscript to PLOS ONE. After careful consideration, we feel that it has merit but does not fully meet PLOS ONE’s publication criteria as it currently stands. Therefore, we invite you to submit a revised version of the manuscript that addresses the points raised during the review process.

I have some major concerns about your study, that the revision process did not completely clarify. In particular, as stated later, the most serious concerns regard irregularities in the study design--an uncontrolled intervention that introduces variability in subject practice and is not included as a covariate in the analysis, and data analysis that uses only a third of the outcome data with no explanations.

Please try to address all Reviewers comments. In particular, takes in great consideration Reviewer's 2 suggestions, since they reflects the Editor's point of view. 

We look forward to receiving your revised manuscript.

Kind regards,

Laura Pasin

Academic Editor

PLOS ONE

Additional Editor Comments (if provided):

The most serious concerns regard irregularities in the study design--an uncontrolled intervention that introduces variability in subject practice and is not included as a covariate in the analysis, and data analysis that uses only a third of the outcome data with no explanations.

Reviewers' comments:

Reviewer's Responses to Questions

**Comments to the Author**

1. If the authors have adequately addressed your comments raised in a previous round of review and you feel that this manuscript is now acceptable for publication, you may indicate that here to bypass the “Comments to the Author” section, enter your conflict of interest statement in the “Confidential to Editor” section, and submit your "Accept" recommendation.

Reviewer #1: All comments have been addressed

Reviewer #2: (No Response)

2. Is the manuscript technically sound, and do the data support the conclusions?

Reviewer #1: Yes

Reviewer #2: Partly

3. Has the statistical analysis been performed appropriately and rigorously? 

Reviewer #1: Yes

Reviewer #2: Yes

4. Have the authors made all data underlying the findings in their manuscript fully available?

Reviewer #1: Yes

Reviewer #2: Yes

5. Is the manuscript presented in an intelligible fashion and written in standard English?

Reviewer #1: Yes

Reviewer #2: Yes

6. Review Comments to the Author

Reviewer #1: (No Response)

Reviewer #2: Major Comments

1. The authors appear to use the term “proficiency” to mean skill at intubating mannequins. However, I generally associate intubation proficiency with competence in patients, since performing the procedure on mannequins is unimportant and uncommon for attending physicians. Therefore, the authors should define what they mean by proficiency early in the manuscript.

2. Title. The current title would give me the wrong impression about the article because I think about intubation proficiency in terms of outcomes in patients and the paper does not investigate patient outcomes. The first sentence should be more specific, something like “Does seniority always correlate with intubation performance in mannequins?”

4. Lines 35-36. I would be specific in the sentence beginning “However, when devices used . . .” I recommend, “However, VS demonstrated shorter duration than medical students using Glidescope and direct laryngoscopy.”

5. Lines 102-123. What was the purpose of the 90-minute practice phase? I can imagine several explanations: 1) The practice session was offered as optional training and provided an opportunity then to recruit and perform research on a large number of subjects. 2) The practice session was intended to give subjects familiarity with the equipment but was not intended as an intervention. 3) The practice session was an intervention for another study. The methods section should explain the purpose of the practice session and whether it was considered one of the experimental procedures.

6. Lines 110-123. Why did the subjects perform 3 attempts with each modality in each scenario, if only 1 attempt in each was used for analysis. This should be explained in the Methods session.

To be frank, it looks like the experimental design and analysis plan initially involved analyzing the effects of the practice session using all 12 intubation attempts and that this plan was changed after the study was completed.

7. Line 213. The statement that “seniority correlates with success rate but not proficiency” is an incomplete and somewhat inaccurate statement of the key findings. The study does in fact show that attending physicians were more likely than medical students to intubate the mannequin trachea successfully. In addition, attending physicians were faster than medical students on Glidescope and direct laryngoscopy intubations. These are the metrics that I would most closely associate with proficiency at intubating a mannequin. The discussion should reflect this finding. The force measurements do not fit the picture the authors would expect if attending physicians skill exceeded that of junior subjects, i.e. less force exerted by attendings. However, I am unsure of whether force actually correlates with skill at intubating a mannequin and I am unsure whether the authors’ expectation of lower force by senior intubators is correct.

My take on the study is that it measured different metrics that the authors believe are related to skill when intubating a mannequin. Some metrics show a level of skill in the senior subjects that is lacking in the junior subjects, while other metrics (the forces) do not give the result that the authors think is consistent with greater skill in senior subjects compared to juniors.

8. Line 284. The most significant limitation in this study is the failure to control the extent of subjects’ practice in the 90-minute session before the test phase. The practice session constitutes an uncontrolled intervention, even if not intended that way. Subjects practiced until they perceived that they were ready to move on, not in the controlled fashion if practiced were allowed to some pre-set number of replicates or until a standardized performance was reached. The variability in practice could have substantially increased variability among subjects, diminishing the study’s power toward distinguishing differences across junior and senior groups of subjects. The limitations section should acknowledge this potential problem.

If the authors changed the experimental design or analysis plan after the study was complete, they should acknowledge this as a limitation as well.

9. The Cormack Lehane grade view measured by the MW11 system estimates the line of sight view obtained during the intubation attempt. However, operators generally obtain a grade 1 or grade 2 view with the Glidescope and Trachway camera chips, even on difficult airways. Operators have would have no reason optimize the line-of-sight view since they have the video view, and usually pay no attention to the line of sight. Thus, the MW11 measurements of line-of-sight view are irrelevant for intubation with the Glidescope and Trachway and the analysis represented by Table 1 is meaningless. The authors may want to re-think incorporating the Cormack and Lehane view as a measure in this paper.

Minor Comments

10. Title. The term “PGY” will be confusing to the American medical audience because PGY residents are physicians. More specificity might be possible by changing the phrase in the title to “medical students, residents, and attending physicians.”

11. Line 26. I recommend changing the phrase “skill proficiency” to “other skills.”

12. Lines 27 and 80. The authors do not define and I do not understand the term “invisible risks and deficiencies.” I would remove the term and any further discussion because it is not closely related to the goal of evaluating how parameters measured during mannequin intubation vary among individuals with different levels of seniority.

13. Line 55. Degree of success is most importantly evaluated by whether the tube is placed in the trachea. I would re-phrase the sentence to, “Intubation may also be evaluated by . . .”

14. Lines 77-78. The main objective should be changed to “explore differences in intubation success rate and other measurements during intubation of the Kyoto Kagaku MW11: difficult airway management (DAM) simulator evaluation system.” (consistent with item 11 above)

15. Lines 102-110. Did subjects practice on direct laryngoscopy, Trachway, and Glidescope during the 90-minute practice phase? Please specify in the text at this point.

16. Lines 131-132. In item 16 of my last review, I was interested in whether the Kyoto-Kagaku MW.11 system measurement of Cormack and Lehane view had been validated, i.e. was the view accurate when compared to the view reported by a trained operator performing the laryngoscopy. The authors did not answer this question

17. Line 128, Line 33: The proportion of both lungs ventilated is not reported later in manuscript. The term could be deleted line 128 and in the abstract.

18. Line 218. Reference 17 is described as a report of a correlation between seniority and intubation speed, but the paper describes emergency patient intubations and the abstract abstract of this paper does not mention speed.

19. Line 223. Contrary to the text, reference 18 reports that the literature is not consistent on whether videolaryngoscopy hastens or prolongs intubation.

20. Line 269. I don’t think you should to ascribe shortcomings to the senior physicians based on their performance. The 4 mannequin scenarios appear to be quite difficult and the attendings achieved the highest success rate of any of the groups, even though one presumes that they had no training on these scenarios. Other groups generated lower dental forces, but this may have contributed to their lower success rates. I would suggest replacing “shortcomings” with “greater dental force” or something more specific.

21. The phrase, “across seniority,” is repeated in the title to Supplementary Table 1.

7. PLOS authors have the option to publish the peer review history of their article (what does this mean?). If published, this will include your full peer review and any attached files.

Reviewer #1: No

Reviewer #2: No

---

## [Author Response · Author response to Decision Letter 1]

9 Sep 2021

Reviewer #2: 

Reviewer #2: Major Comments

1. The authors appear to use the term “proficiency” to mean skill at intubating mannequins. However, I generally associate intubation proficiency with competence in patients, since performing the procedure on mannequins is unimportant and uncommon for attending physicians. Therefore, the authors should define what they mean by proficiency early in the manuscript.

Our Response:

We thank the reviewer for this comment. The term proficient intubators refers to providers who can adjust their techniques to accommodate the conditions of patients to maintain an airway and provide ventilation quickly [1]. The most proficient intubators are associated with fast diagnoses, quality intubations, and fewer incidents [2-3]. Although previous study has assessed physicians’ intubation proficiency on a simulator [4], we agreed that it was better to illustrate the findings of the current study with the intubation performance as described in most of the existing research (e.g. [5]). We have now revised our research focus to intubation performance instead of intubation proficiency.

1. Soar, J., et al., European resuscitation council guidelines for resuscitation 2015: section 3. Adult advanced life support. Resuscitation, 2015. 95: p. 100-147.

2. Bernhard, M., et al., Developing the skill of endotracheal intubation: implication for emergency medicine. Acta Anaesthesiologica Scandinavica, 2012. 56(2): p. 164-171.

3. Higgs, A., et al., Guidelines for the management of tracheal intubation in critically ill adults. British journal of anaesthesia, 2018. 120(2): p. 323-352.

4. Gillett, B., et al., Skill proficiency is predicted by intubation frequency of emergency medicine attending physicians. Western Journal of Emergency Medicine 20.4 (2019): 601.

5. Kannaujia, A., et al., Comparative evaluation of intubation performances using two different barrier devices used in the COVID-19 era: A manikin based pilot study. Saudi Journal of Anaesthesia, 15(2), p.86.

2. Title. The current title would give me the wrong impression about the article because I think about intubation proficiency in terms of outcomes in patients and the paper does not investigate patient outcomes. The first sentence should be more specific, something like “Does seniority always correlate with intubation performance in mannequins?”

Our Response:

Thank you, we have now revised our title to be more explicit as:

“Does seniority always correlate with simulated intubation performance? Comparing endotracheal intubation performance across medical students, residents, and physicians using a high-fidelity simulator.”

4. Lines 35-36. I would be specific in the sentence beginning “However, when devices used . . .” I recommend, “However, VS demonstrated shorter duration than medical students using Glidescope and direct laryngoscopy.”

Our Response:

We have revised our result to be more specific as follows:

“However, VS demonstrated shorter duration than medical students using Glidescope and direct laryngoscopy, whereas VS and R applied significantly more force on the incisor in the normal airway and rigid neck scenario respectively.”

5. Lines 102-123. What was the purpose of the 90-minute practice phase? I can imagine several explanations: 1) The practice session was offered as optional training and provided an opportunity then to recruit and perform research on a large number of subjects. 2) The practice session was intended to give subjects familiarity with the equipment but was not intended as an intervention. 3) The practice session was an intervention for another study. The methods section should explain the purpose of the practice session and whether it was considered one of the experimental procedures.

Our Response:

The 90-minute training section phase comprised 3 sections including material review (30 minutes), technique familiarity (30 minutes), and manikin practice (30 minutes). We have presented this in Figure 1, and clarified it in the method section as follows:

“Each session lasted about 90 minutes and comprised 3 sections: 30-minute material review, 30-minute device familiarization (i.e., practice on direct laryngoscopy, Trachway®, and Glidescope®), and 30-minute manikin practice section (Fig 1). This design offered sufficient demonstration and practice time for learners to become familiar with the devices and intubation on the manikin but was not intended as an intervention.”

As the reviewer mentioned, the practice session was intended to give subjects familiarity with the equipment but was not intended as an intervention.

6. Lines 110-123. Why did the subjects perform 3 attempts with each modality in each scenario, if only 1 attempt in each was used for analysis. This should be explained in the Methods session.

To be frank, it looks like the experimental design and analysis plan initially involved analyzing the effects of the practice session using all 12 intubation attempts and that this plan was changed after the study was completed.

Our Response:

We have analyzed all 12 intubations (4 scenarios and 3 devices) as described in the section on statistical analysis. Consequently, each subject was required to attempt each modality at least once in each scenario since only the first attempt would be taken into consideration. Due to our inability to anticipate the effect of practice during assessment, we do concede that learners who attempted multiple times during assessment improvement may have been a consequence of their practice. It has been included in our limitations as described in the previous response to reviewers. In Line 295 – 298, we described: 

“Moreover, we did not control the learners’ number of attempts during the practice section and assessment; we only analyzed their first attempt at each scenario. Learners with a higher number of attempts may have benefitted from a practice effect and thereby improved their performance in later scenarios.”

7. Line 213. The statement that “seniority correlates with success rate but not proficiency” is an incomplete and somewhat inaccurate statement of the key findings. The study does in fact show that attending physicians were more likely than medical students to intubate the mannequin trachea successfully. In addition, attending physicians were faster than medical students on Glidescope and direct laryngoscopy intubations. These are the metrics that I would most closely associate with proficiency at intubating a mannequin. The discussion should reflect this finding. The force measurements do not fit the picture the authors would expect if attending physicians’ skill exceeded that of junior subjects, i.e. less force exerted by attendings. However, I am unsure of whether force actually correlates with skill at intubating a mannequin and I am unsure whether the authors’ expectation of lower force by senior intubators is correct.

My take on the study is that it measured different metrics that the authors believe are related to skill when intubating a mannequin. Some metrics show a level of skill in the senior subjects that is lacking in the junior subjects, while other metrics (the forces) do not give the result that the authors think is consistent with greater skill in senior subjects compared to juniors.

Our Response:

We appreciate the concise and powerful summary of the metrics of performance in intubating manikins of varying seniority. As noted in #1, our study has been revised to focus on intubation performance instead. This discussion section begins with the main effect (i.e., the primary finding) of seniority, and then discusses the significant interaction effect later. 

We have revised the discussion paragraph to include a more detailed discussion of the force issue on soft tissue as follows:

“As we know, the glossopharyngeal nerve is positioned superior to the anterior surface of the epiglottis, and the vagus nerve has sensory pathways running distally from the posterior epiglottis to the lower airway. The pressure of laryngoscopy during endotracheal intubation will stimulate both nerves and may cause lethal cardiovascular responses [1, 2]. Therefore, minimizing the applied laryngoscopy force could reduce the haemodynamic response, as well as the local tissue trauma associated with intubation [3]. Accordingly, the laryngoscopy force applied to the soft tissues of the oral mucosa should be reduced as the physician's seniority and competence in the quality of their care increases.”

1. Kovac AL. Controlling the hemodynamic response to laryngoscopy and endotracheal intubation. J Clin Anesth. 1996 Feb;8(1):63-79. doi: 10.1016/0952-8180(95)00147-6.

2. Akbar AN, Muzi M, Lopatka CW, Ebert TJ. Neurocirculatory responses to intubation with either an endotracheal tube or laryngeal mask airway in humans. J Clin Anesth. 1996 May;8(3):194-7. doi: 10.1016/0952-8180(95)00228-6.

3. Russell T, Khan S, Elman J, Katznelson R, Cooper RM. Measurement of forces applied during Macintosh direct laryngoscopy compared with GlideScope® videolaryngoscopy. Anaesthesia. 2012 Jun;67(6):626-31. doi: 10.1111/j.1365-2044.2012.07087.x.

8. Line 284. The most significant limitation in this study is the failure to control the extent of subjects’ practice in the 90-minute session before the test phase. The practice session constitutes an uncontrolled intervention, even if not intended that way. Subjects practiced until they perceived that they were ready to move on, not in the controlled fashion if practiced were allowed to some pre-set number of replicates or until a standardized performance was reached. The variability in practice could have substantially increased variability among subjects, diminishing the study’s power toward distinguishing differences across junior and senior groups of subjects. The limitations section should acknowledge this potential problem.

If the authors changed the experimental design or analysis plan after the study was complete, they should acknowledge this as a limitation as well.

Our Response:

The term “practice time” refers to the number of attempts during the practice section instead of time for practice. The time for practice before assessment was controlled, with each learner allowed to practice for 30 minutes as mentioned before (refer to #5). However, the number of attempts during the 30-minute practice section were not manipulated. Therefore, we didn’t record their intubation performance and frequency during the practice section. We have now rephrased the misleading term “practice time”. We agree that the number of attempts may have affected learners’ later performance, and have now addressed the problem as a limitation more explicitly. 

9. The Cormack Lehane grade view measured by the MW11 system estimates the line of sight view obtained during the intubation attempt. However, operators generally obtain a grade 1 or grade 2 view with the Glidescope and Trachway camera chips, even on difficult airways. Operators have would have no reason optimize the line-of-sight view since they have the video view, and usually pay no attention to the line of sight. Thus, the MW11 measurements of line-of-sight view are irrelevant for intubation with the Glidescope and Trachway and the analysis represented by Table 1 is meaningless. The authors may want to re-think incorporating the Cormack and Lehane view as a measure in this paper.

Our Response:

Thank you for your comments. We have reconsidered the significance of Cormack Lehane grade performance of various devices in this study. As the reviewer stated, the line-of-sight view from a video laryngoscope (both Glidescope and Trachway) is from the camera lens of the video chipset, just like the view of an operator lying in front of the throat. Cormack Lehane grade differs from the original grading concept. Therefore, we have removed the part of the analysis involving both video laryngoscopes and re-presented the results.

10. Title. The term “PGY” will be confusing to the American medical audience because PGY residents are physicians. More specificity might be possible by changing the phrase in the title to “medical students, residents, and attending physicians.”

Our Response:

Thank you - we agree this is a much clearer title for the general audience and have revised it in the latest version.

11. Line 26. I recommend changing the phrase “skill proficiency” to “other skills.”

Our Response:

We have revised this phrase in line 26, and line 76 also. Thank you.

12. Lines 27 and 80. The authors do not define and I do not understand the term “invisible risks and deficiencies.” I would remove the term and any further discussion because it is not closely related to the goal of evaluating how parameters measured during mannequin intubation vary among individuals with different levels of seniority.

Our Response:

We have now deleted this term and the further discussion.

13. Line 55. Degree of success is most importantly evaluated by whether the tube is placed in the trachea. I would re-phrase the sentence to, “Intubation may also be evaluated by . . .”

Our Response:

Thank you for your comments, and we have now rephrased as follows: 

“The degree of success of intubation is traditionally evaluated by the time required and the achievement of both-lung ventilation. Intubation may also be evaluated by additional indicators such as applied force and glottic view that could affect patient outcome.”

14. Lines 77-78. The main objective should be changed to “explore differences in intubation success rate and other measurements during intubation of the Kyoto Kagaku MW11: difficult airway management (DAM) simulator evaluation system.” (consistent with item 11 above)

Our Response:

Thank you, and we have revised it.

15. Lines 102-110. Did subjects practice on direct laryngoscopy, Trachway, and Glidescope during the 90-minute practice phase? Please specify in the text at this point.

Our Response:

Thank you. We have now specified these three devices: 

“Each session lasted about 90 minutes and comprised 3 sections: 30-minute material review, 30-minute device familiarization (i.e., practice on direct laryngoscopy, Trachway®, and Glidescope®), and 30-minute manikin practice section (Fig 1).”

16. Lines 131-132. In item 16 of my last review, I was interested in whether the Kyoto-Kagaku MW.11 system measurement of Cormack and Lehane view had been validated, i.e. was the view accurate when compared to the view reported by a trained operator performing the laryngoscopy. The authors did not answer this question

Our Response:

The Kyoto-Kagaku MW.11 difficult airway management simulator was developed at Waseda University in 2006 to serve as an active training device and as an advanced evaluation tool for surgical instruments and medical procedures [1, 2]. The simulator was built based on the concepts of a Japanese anesthesia textbook (Tracheal Intubation Visual Manual of Clinical Basic Techniques) [3]. However, we were unable to discover any English literature describing the process of Cormack and Lehane grade verification [4]. This has been addressed as a study limitation.

1. Y. Noh et al., Development of the airway management training system WKA-4: For improved high-fidelity reproduction of real patient conditions, and improved tongue and mandible mechanisms. 2011 IEEE International Conference on Robotics and Automation. 2011:1726-31.

2. Y. Noh et al., Development of the airway Management Training System WKA-5: Improvement of mechanical designs for high-fidelity patient simulation. 2012 IEEE International Conference on Robotics and Biomimetics (ROBIO). 2012:1224-9.

3. Kazuyoshi Aoyama(2009). Tracheal Intubation Visual Manual of Clinical Basic Techniques. Japan：Yodosha.

4. I. Okuyama et al., Reproduction and verification of airway difficulties using airway management training system WKA-5. Journal of Japan Society Computer Aided Surgery.Vol13. No3. 2011.P224-225.

17. Line 128, Line 33: The proportion of both lungs ventilated is not reported later in manuscript. The term could be deleted line 128 and in the abstract.

Our Response:

Thank you, we have now deleted this parameter. 

18. Line 218. Reference 17 is described as a report of a correlation between seniority and intubation speed, but the paper describes emergency patient intubations and the abstract of this paper does not mention speed.

Our Response:

Thank you, we have updated our reference:

Kim, Sin Young, et al. "How much experience do rescuers require to achieve successful tracheal intubation during cardiopulmonary resuscitation?." Resuscitation 133 (2018): 187-192.

19. Line 223. Contrary to the text, reference 18 reports that the literature is not consistent on whether video-laryngoscopy hastens or prolongs intubation.

Our Response:

Thank you, we have updated our reference:

Griesdale, Donald EG, et al. "Glidescope® video-laryngoscopy versus direct laryngoscopy for endotracheal intubation: a systematic review and meta-analysis." Canadian Journal of Anesthesia/Journal canadien d'anesthésie 59.1 (2012): 41-52.

20. Line 269. I don’t think you should to ascribe shortcomings to the senior physicians based on their performance. The 4 mannequin scenarios appear to be quite difficult and the attendings achieved the highest success rate of any of the groups, even though one presumes that they had no training on these scenarios. Other groups generated lower dental forces, but this may have contributed to their lower success rates. I would suggest replacing “shortcomings” with “greater dental force” or something more specific.

Our Response:

Thank you, we have rephrased the term to be more explicit.

21. The phrase, “across seniority,” is repeated in the title to Supplementary Table 

Our Response:

Corrected, thank you.

---

## [Editor Report · Decision Letter 2]

15 Sep 2021

Does seniority always correlate with simulated intubation performance? Comparing endotracheal intubation performance across medical students, residents, and physicians using a high-fidelity simulator.

PONE-D-21-11478R2

Dear Dr. Liao,

We’re pleased to inform you that your manuscript has been judged scientifically suitable for publication and will be formally accepted for publication once it meets all outstanding technical requirements.

Kind regards,

Laura Pasin

Academic Editor

PLOS ONE
---

## [Editor Report · Acceptance letter]

17 Sep 2021

PONE-D-21-11478R2 

Does seniority always correlate with simulated intubation performance? Comparing endotracheal intubation performance across medical students, residents, and physicians using a high-fidelity simulator. 

Dear Dr. Liao:

I'm pleased to inform you that your manuscript has been deemed suitable for publication in PLOS ONE. Congratulations! Your manuscript is now with our production department. 

Kind regards, 

on behalf of

Dr. Laura Pasin 

Academic Editor

PLOS ONE